New in vitro system to predict chemotherapeutic efficacy of drug combinations in fresh tumor samples

Kischkel Frank Christian frank.kischkel@therapyselect.de 1
Eich Julia 1
Meyer Carina I. 1
Weidemüller Paula 1
Krapfl Jens 1
Yassin-Kelepir Rauaa 1
Job Laura 1
Fraefel Marius 1
Braicu Ioana 2
Kopp-Schneider Annette 3
Sehouli Jalid 2
De Wilde Rudy Leon 4
1 TherapySelect , Heidelberg , Germany
2 Gynecology Department, Charité Berlin , Virchow Campus Berlin , Germany
3 Division of Biostatistics, German Cancer Research Center , Heidelberg , Germany
4 University Hospital for Gynecology , Carl von Ossietzky University Oldenburg , Germany
Patnaik Santosh
Electronic publication date: 2017 Mar 2
Publication date: 2017
Volume: 5
Electronic Location ID: e3030
Received 2016 Nov 3; Accepted 2017 Jan 25
Copyright: ©2017 Kischkel et al.
Copyright year: 2017
Copyright holder: Kischkel et al.
License: This is an open access article distributed under the terms of the Creative Commons Attribution License, which permits unrestricted use, distribution, reproduction and adaptation in any medium and for any purpose provided that it is properly attributed. For attribution, the original author(s), title, publication source (PeerJ) and either DOI or URL of the article must be cited.
License URL: https://creativecommons.org/licenses/by/4.0/

Keywords: Chemotherapy-resistance-test, Drug combination, Ovarian carcinoma, Drug combination testing, Additive effect, Drug sensitivity testing, Individualized cancer therapy

Funding: German Ministry of Education and Research 0315778A German Ministry of Economic Affairs and Energy EP131017 This work was sponsored by grants from the German Ministry of Education and Research (grant number 0315778A) and the German Ministry of Economic Affairs and Energy (grant number EP131017) and by resources of TherapySelect. There was no additional external funding received for this study. The funders had no role in study design, data collection and analysis, decision to publish, or preparation of the manuscript.

==============================
Background

To find the best individual chemotherapy for cancer patients, the efficacy of different chemotherapeutic drugs can be predicted by pretesting tumor samples in vitro via the chemotherapy-resistance (CTR)-Test®. Although drug combinations are widely used among cancer therapy, so far only single drugs are tested by this and other tests. However, several first line chemotherapies are combining two or more chemotherapeutics, leading to the necessity of drug combination testing methods.

Methods

We established a system to measure and predict the efficacy of chemotherapeutic drug combinations with the help of the Loewe additivity concept in combination with the CTR-test. A combination is measured by using half of the monotherapy’s concentration of both drugs simultaneously. With this method, the efficacy of a combination can also be calculated based on single drug measurements.

Results

The established system was tested on a data set of ovarian carcinoma samples using the combination carboplatin and paclitaxel and confirmed by using other tumor species and chemotherapeutics. Comparing the measured and the calculated values of the combination testings revealed a high correlation. Additionally, in 70% of the cases the measured and the calculated values lead to the same chemotherapeutic resistance category of the tumor.

Conclusion

Our data suggest that the best drug combination consists of the most efficient single drugs and the worst drug combination of the least efficient single drugs. Our results showed that single measurements are sufficient to predict combinations in specific cases but there are exceptions in which it is necessary to measure combinations, which is possible with the presented system.

Introduction

Progressively more is known about the underlying causes for the formation of a tumor, which are highly complex and involve in general several mechanisms at the molecular level. Therefore, tumors are heterogeneous and every individual patient has a specific response profile to the selected treatment concerning the varying chemosensitivity of tumor cells (Blom et al., 2016). However, the standard therapy for cancer treatment is based on an average response of a group of patients with similar tumor types which leads to a lack of benefit in several cases and to the necessity of a next-line therapy (Blom et al., 2016). One method to improve the therapeutic outcome is to priory assess the resistance of tumors from individual patients to chemotherapeutic drugs in order to provide the most efficient therapy to every single patient (Nygren & Larsson, 2008). Such therapy selection strategies are considered to have great potential to advance cancer treatment (Pusztai et al., 2004; Ludwig & Weinstein, 2005; Ioannidis, 2007; Trusheim, Berndt & Douglas, 2007). A recent prospective clinical trial showed the benefit of using an in vitro chemotherapeutic test in ovarian cancer (Rutherford et al., 2013; Grendys et al., 2014). This relationship can be intuitively explained by the assumption that if a drug is ineffective in a simple system like an in vitro test using isolated tumor cells, the probability that is has an effect in a patient is highly unlikely (Nygren et al., 1994). One option of such an in vitro test is the so-called Chemotherapy Resistance Test (CTR-Test®) which was the chosen method in this paper. The CTR-Test is identical to a formerly described extreme drug resistance (EDR) assay (Kern & Weisenthal, 1990). EDR assays are applied to identify chemotherapeutics which are ineffective rather than to find chemotherapeutics which are likely to show an effect. Thereby, the treatment of a patient with a toxic agent that does not result in a therapeutic benefit can be prevented (Tattersall & Harnett, 1986; Myers et al., 1987; Beck, 1987). It is known that the capability to predict drug resistance is presumably >95% whereas the ability to predict chemo-sensitivity lies around 60% (Kim et al., 2009). The CTR-Test shows a >99% accuracy in finding ineffective chemotherapeutics that do not produce a clinical response (Kern & Weisenthal, 1990).

So far, only single drugs are tested in this system. However, in the case of ovarian cancer the standard first-line treatment is a combination therapy of carboplatin together with paclitaxel (Du Bois et al., 2003; Pfisterer et al., 2006; Du Bois et al., 2006; Bookman et al., 2009). In clinical practice, combination therapies are more frequently applied and in general the benefits of a combination therapy are reduced side effects as well as reduced drug resistance. The reason for reduced side effects is that lower doses of the two drugs can be applied which still lead to the same efficacy as a higher dose of the particular monotherapy but avoid toxicity. Reduced drug resistance is achieved by diverse mechanisms of action of the two chemotherapeutics (Sparano, 1999; Prisant, 2002; Tallarida, 2006; Kashif et al., 2015).

There are several paper published showing that it is sufficient to test single drugs via the CTR-Test and use their efficacy data to find effective combination therapies, which lead to a clinical response (Mehta et al., 2001; Holloway et al., 2002; Loizzi et al., 2003; d’Amato et al., 2009; Kim et al., 2009; Matsuo et al., 2009). The question arises whether in general therewould be an improvement in cancer therapy when combinations instead of single drugs are tested or if the testing of single drugs is sufficient for a good clinical prediction. To our knowledge no effective in vitro test system or test principle for testing drug combinations exists. Therefore, there is need of an enhanced in vitro diagnostic test system which enables the clinically relevant investigation of the efficacy of drug combinations. In this paper, we used a new system to test drug combinations in vitro with the CTR-Test.

Material and Methods

Tumor tissue samples collection

The tumor specimen were collected as part of the commercially offered CTR-Test or as part of a clinical trial. All included samples were left over after the commercial or clinical trial assay was performed. For all samples patient’s consent forms exist, which allow further scientific investigations. In total 273 ovarian carcinoma, one malignant melanoma, one small cell lung cancer, one mamma carcinoma, four colon carcinoma and one NSCLC biopsies were collected. After surgery tissue samples were directly stored in medium and sent to the laboratory (TherapySelect, Heidelberg, Germany) to perform the CTR-Test. The specimens arrived at TherapySelect within 24 h and were processed on the same day.

CTR-Test

The tissues were processed and the CTR-Test was performed by TherapySelect, according to a published protocol (Kern & Weisenthal, 1990; d’Amato et al., 2009). Briefly, fresh tumor material is minced into single cells and small cellular aggregates (spheroids). Viability and percentage of tumor cells is determined by an external pathology. Cells are seeded in a culture dish, in which they cannot adhere, and directly treated with a specific chemotherapeutic drug or a drug combination. After 72 h incubation tritiated thymidine (H3-Thymidine) is added to the cells. After additional 48 h cells are harvested onto glass fiber filters and the isotope uptake into the DNA is analyzed by scintillation counting. The data obtained are counts per minute (cpm). Cells cultured without drugs are used as negative control and cells treated with a lethal dose are the positive control. The chemotherapeutic effect is measured in percent cell growth inhibition (PCI) using the formula: PCI = (cpm(treated cells) − cpm(positive control))/(cpm(negative control) − cpm(positive control)).

Drugs used for analysis

All drugs used in this study were selected for therapeutic relevance and were validated for the CTR-Test before the analysis of the tumor samples for this study. For the validation, various drug concentrations were tested in the CTR-Test with freshly isolated tumor samples in order to find a concentration which shows a sufficient distribution of drug action among the tumor samples. Final applied drug concentrations are presented in Table 1. For the measurements of drug combinations (two drugs) half of the concentration of each single drug was used to treat the cells simultaneously.

Table 1 Used chemotherapeutics and their concentration.

Name	Chemical class/mechanism of action	Used monotherapy concentration (µg/ml)	Used combination concentration (µg/ml)	
5-Fluorouracil	Thymidylate synthase inhibitor/antimetabolites	3.0	1.5	
Carboplatin	Platinum-based antineoplastic agent/DNA interaction and interference with DNA repair	3.81	1.905	
Caelyx® (Doxorubicin —liposomal)	Intercalating DNA/anthracycline antitumor antibiotic	3.62	1.81	
Docetaxel	Interference in cell division	1.94	0.97	
Doxorubicin	Intercalating DNA/anthracycline antitumor antibiotic	0.1	0.05	
Etoposide	Topoisomerase inhibitor	3.62	1.81	
Gemcitabine	Nucleoside analog	0.014	0.007	
Oxaliplatin	Platinum-based antineoplastic agent/ DNA interaction and interference with DNA repair	1	0.5	
Paclitaxel	Interference in cell division	2.1	1.05	
Topotecan	Topoisomerase inhibitor	0.1	0.05	
SN-38	Antineoplastic drug/inhibition of topoisomerase 1	0.012	0.006	

Statistical analysis

PCI values were obtained on the response rate of a tumor sample collective to a certain concentration of a chemotherapeutic drug. Frequency distributions of PCI values were generated by joining the mid-points of 4- or 5-bin histograms by a smooth curve in Excel.

The frequency distributions were applied to identify the three resistance categories SR, MR and ER. Therefore, the mean (μ) and the standard deviation (SD) were determined. Mean values are presented in the corresponding figures, SD values can be found in Table  S1. ER is characterized as PCI <μ − 1 SD, MR as PCI > ER but <μ and SR as PCI ≥μ.

We compared the measured and calculated PCI values of the different combinations by determining the Pearson correlation coefficient. To assess agreement between calculated and measured values we showed the difference of calculated and measured values vs. the average of both values in a Bland-Altman Plot (Fig. S1)

Human studies

TherapySelect offers the commercial testing of drug efficacies for viable tumor samples. For this testing, viable tumor specimen is shipped to TherapySelect’s laboratory. Customers (patients) fill out a consent and order form. In this form there is a section in which the patients can choose whether left over material can be used for research purposes. For all used samples patient’s consent forms exist, which allow further scientific investigations. For this paper no ethical approval was requested, since human tissue was initially removed for commercially performed diagnostic purposes.

Results

New system to measure and calculate efficacy of drug combinations and determination of the correlation

The response rate of a collective of tumor samples to a certain concentration of a chemotherapeutic drug experiences a standard distribution. An ideal testing concentration of a drug is found if the histogram spans over the full spectrum of percent cell growth inhibition (PCI) and the curve’s mean overlaps with 50% inhibition effect. Fig. 1A shows an ideal distribution with the perfect concentration. This concept is used to find and validate concentrations for single drug therapy testing. The chosen concentration (Table 1) is in a physiological range, that means close to or below the maximal serum concentration for the individual drugs.

Figure 1 New system for testing drug combinations in vitro with the CTR-Test.

(A) Ideal distribution of percent cell growth inhibition (PCI) values from a tumor patient collective using the ideal concentration of a drug (black curve, mean at 50% PCI). Two individually measured drugs A and B at ideal concentrations show identical ideal distribution curves (black curve). One half of the ideal concentration of drug A can be replaced by one half of the ideal concentration of drug B or vice versa when the drugs are applied in combination. This results in a curve identical to single curves of A or B (black curve, additive effect). A curve shift to the left would be due to an antagonistic effect, a shift to the right would be due to a synergistic effect when two drugs are combined (grey dashed curves). (B) In an ideal situation the measured PCI values are equal to the calculated PCI values (formula see text) of a drug combination. The ideal PCI values plotted against each other result in a correlation coefficient of 1 (additive effect). Data points above or below the line show an antagonistic or synergistic effect, respectively. (C) 273 ovarian carcinoma samples (99 primary, 140 recurrent and 34 unknown ovarian carcinoma cases) were treated with carboplatin and paclitaxel alone or in combination. PCI values of the single drugs were determined with the CTR-test. The PCI values of the combination were measured with the CTR-test as well as calculated via the presented formula. For measuring the combination, half of the concentration of each drug was used. The frequency distributions of PCI values of the different settings were plotted (black dashed curve: carboplatin alone, grey dashed curve: paclitaxel alone, grey curve: measured PCI of combination, black curve: calculated PCI of combination). The mean PCI value of all curves spreads around 70%. (D) The measured and calculated PCI values of the combination were plotted against each other resulting in a correlation coefficient of 0.84 (Table 2).

In case of combination testing the individual concentrations have to be adapted. Therefore, the concept of Loewe additivity was customized to this question. The underlying assumption of the additivity model is that two inhibitors operate through similar mechanisms on a target and dose substitution is the following consequence (Loewe, 1953; Berenbaum, 1989). Adapting this concept to our issue, in an ideal case the distribution curves of separately measured drugs, A and B, used at the ideal concentration reveal two identical response histograms. Therefore, one half of the ideal concentration of drug A could replace one half of the ideal concentration of drug B or vice versa. This would lead to a histogram identical to a curve produced by the full concentration of drug A or B, respectively.

With this in mind the extent of inhibition in combination (PCI(a + b)) can be calculated by a mathematical model based on the inhibition effects (PCIa or PCIb) of the mono-drugs (A or B) (Fig. 1A) (1) PCI(a + b)=PCIa2+PCIb2.

The inhibition effects of the two mono-drugs (PCIa and PCIb) are divided by two because half of the concentration of drug A and drug B would be used if they would be applied in combination. Both values are added and result in the inhibition effect of the combination (PCI(a + b)).

In an ideal setting the value of the measured combination effect is equal to the calculated effect. In an ideal plot of measured versus calculated effect, the correlation coefficient would be 1 and is linear correlated, which represents the additive effect (Fig. 1B). In this ideal setting, a data point either above or below the line would stand for an antagonistic or a synergistic effect, respectively. In the curve representation of the patients collective, those effects are seen by a shift to the left or to the right of the histogram in case of antagonism or synergy, respectively (Fig. 1A). Synergy can be defined as a stronger cell growth inhibition effect measured than the calculated combination effect. Antagonistic would mean a reduced effect in combination (Chou, 2010; Kashif et al., 2014).

To test this theory, 273 ovarian carcinoma samples (99 primary, 140 recurrent and 34 unknown ovarian carcinoma cases) were treated with carboplatin and paclitaxel alone or in combination. For the combination measurement half of the concentration of each tested mono-drug (carboplatin/paclitaxel) was applied. Cell growth inhibition effects were measured with the CTR-Test and histograms were created. Additionally, the cell growth inhibition effect was calculated for the combination by using the presented formula. The four different distributions of effect in the tumor sample collective are presented in Fig. 1C. The curve of carboplatin is slightly shifted compared to the histogram of paclitaxel. However, the calculated curve lies in between the single drug histograms and the calculated curve is in general more narrow. As mentioned before, the chosen chemotherapeutic concentration should lead to a mean at a PCI value of 50% in the histogram of the collective. However, all curves have a mean which spreads around 70% (Fig. 1C). The calculated mean 69.3% of the combination lays between the mean of carboplatin and paclitaxel with 67.7% and 70.9%, respectively. The measured value lays above the calculated mean with 73.0%. To further analyze the accuracy of predicting the combination effect, the calculated and measured PCI values are plotted against each other (Fig. 1D). The combination of carboplatin and paclitaxel leads to a correlation of 0.84 (Table 2).

Table 2 Correlation coefficient (R) and resistance classification (calculated equal to measured values) of carboplatin combinations.

	Carboplatin & Paclitaxel	Carboplatin & Caelyx®	Carboplatin & Doxorubicin	Carboplatin & Etoposide	Carboplatin & Topotecan	Carboplatin & Docetaxel	Carboplatin & Gemcitabine	
Correlation coefficient (R)	0.84	0.84	0.85	0.83	0.82	0.68	0.34	
Resistance classification: calculated = measured	77.0%	72.5%	71.0%	68.8%	74.2%	64.3%	48.7%	
Data set size (n)	273	39	29	32	29	28	39	

Evaluation of predicting efficacy by measured drug combinations versus the calculated on basis of single drug measurement

To classify the chemoresistance of tumors to certain drugs, the response histogram of a patient collective is used to define three resistance categories (Mehta et al., 2001; Holloway et al., 2002; Loizzi et al., 2003; Kim et al., 2009; Matsuo et al., 2009). Extreme resistance (ER) is marked by PCI < Mean − 1 SD (standard deviation). PCI > ER but <Mean values are classified as medium resistance (MR). All PCI ≥ Mean values are called slight resistance (SR) (Fig. 2A). Mean values used for determination of resistance categories are presented in the corresponding figures, the standard deviations can be found in Table S1. Those resistant classifications are used to predict treatment success by the use of a monotherapy for the individual patient (Kern & Weisenthal, 1990).

Figure 2 Evaluation of predicting efficacy of the new system for chemoresistance classification.

(A) The distribution of PCI values of a tumor patient collective for a certain drug applied at a specific concentration is used to classify three chemoresistance categories. ER (extreme resistance) is characterized as PCI < μ (mean) − 1 SD (standard deviation) (dashed red line). PCI > ER but < μ (dashed green line) is classified as MR (medium resistance), PCI ≥μ is classified as SR (slight resistance). (B and C) Data set contains 273 ovarian carcinoma samples. (B) The resistance categories of this data set for carboplatin alone, paclitaxel alone, measured and calculated combination were defined by the system described in (A) and the classification borders for measured and calculated are marked by a green (μ) and red line (μ − 1 SD). The underlying curves are presented in Fig. 1C. These measured and calculated PCI values of the combination carboplatin and paclitaxel are plotted against each other. Individual data points are color-coded depending on the chemoresistance category of the patient for the two single drugs. (C) Single drug resistance categories are compared to the measured categories of the combination carboplatin and paclitaxel.

The influence of the single drug resistance in the combination therapy was analyzed. Therefore, the four histograms of carboplatin, paclitaxel, measured and calculated values, were used to define the PCI values of the resistance borders. In the comparison plot of measured versus calculated PCI, lines are drawn at the borders between ER and MR (red) and between MR and SR (green) for calculated and measured individually. As an example how the resistance borders were defined, it is shown how the resistance borders for the measured combination (carboplatin and paclitaxel) were determined. The Mean value (μ) is 73.3% (see Fig. 1) and indicates the border between SR and MR (green line). The standard deviation (SD) is 14.9% (see Table S1) and therefore the border between MR and ER (μ − 1 SD) lies at 58.4% (red line). The three resulting squares along the diagonal represent overlapping resistance profiles, gained by the measured and the calculated resistance classification (Fig. 2B). In this data set, in 77% of the cases the resistance classification of the measured resistance is equal to the calculated resistance (Table 2).

Additionally, each data point was categorized by the underlying resistance classes of the single drug measurements (Fig. 2C). Comparing single drug resistance categories to the measured combination-categories reveals that if carboplatin and paclitaxel are identical categorized either as SR or ER, the measured resistance category stays in 95% or 100% of the cases SR or ER, respectively. A similar tendency is seen in the case of medium resistance for carboplatin and paclitaxel. Here 75% of the measured samples are categorized as MR. However, for combinations of different single resistance categories the prediction of the resistance profile for the measured combination is less accurate. For the far apart combinations ER with SR there is almost no prediction possible. For combinations closer together, like ER with MR and SR with MR, a tendency can be seen (Fig. 2C).

Applying new system to other drug combinations

The first line standard therapy for treating ovarian carcinoma is a combination therapy of carboplatin together with paclitaxel. We could show that our new approach to use the CTR-Test system for testing drug combinations is functional in this scenario. Most of the measured combination PCI values are in agreement with the calculated ones, basing on the single measurements. As mentioned above, for the relevant resistance categories SR and ER almost all measured values coincide with the calculated values and belong to the correct resistance category. Besides the combination carboplatin and paclitaxel also other chemotherapeutics can be applied to treat ovarian carcinoma. Therefore, we tested carboplatin together with six other chemotherapeutics in the CTR-Test in order to investigate if the new system is also functional in testing other carboplatin combinations. The six chemotherapeutics were: Caelyx® (doxorubicin-hydrochloride in a pegylated liposomal formulation), doxorubicin, docetaxel, etoposide, gemcitabine and topotecan. Cell growth inhibition was determined by the CTR-Test. Due to tumor material limitations only a subset of the 273 ovarian tumor samples was tested with the six carboplatin combinations. 39 ovarian tumor samples were measured for the combination carboplatin-Caelyx (Figs. 3A and 3E), 29 for carboplatin-doxorubicin (Figs. 3B and 3F), 30 for carboplatin-docetaxel (Figs. 4A and 4C), 32 for carboplatin-etoposide (Figs. 3C and 3G), 36 for carboplatin-gemcitabine (Figs. 4B and 4D) and 29 for carboplatin-topotecan (Figs. 3D and 3H). Histograms for the single drugs and in combination were determined and the combined PCI value was calculated using the formula presented above. The histograms of the combinations carboplatin-Caelyx, carboplatin-doxorubicin, carboplatin-etoposide and carboplatin-topotecan show a good distribution of the different PCI curves for the single drugs, the calculated and the measured values (Figs. 3A–3D). However, the histograms for the combinations carboplatin-docetaxel and carboplatin-gemcitabine exhibit a distribution which is divergent in a great extent from an ideal distribution (Figs. 4A and 4B).

The measured and calculated PCI values were plotted against each other to investigate how precise the calculated PCI of the combination was determined. In accordance with the histograms, the data points for the combinations carboplatin-Caelyx, carboplatin-doxurubicin, carboplatin-etoposide and carboplatin-topotecan spread around a regression line and show a similar distribution as the combination carboplatin-paclitaxel (Figs. 3E–3H). The data points for the combinations carboplatin-docetaxel and carboplatin-gemcitabine show a different pattern (Figs. 4C and 4D). In Table 2 the correlation coefficients R as well as the accuracy of resistance classification of the calculated versus the measured values is illustrated.

Figure 3 New system is used to test other drug combinations.

(A–D) 39 ovarian carcinoma samples were treated with carboplatin and Caelyx (A), 29 with carboplatin and doxorubicin (B), 32 with carboplatin and etoposide (C) and 29 with carboplatin and topotecan (D) alone or in combination. PCI of the single drugs and the different combinations was measured with the CTR-Test. In addition, the PCI of the combinations was calculated with the presented formula (1). The frequency distributions of PCI values of the different settings were plotted (black dashed curves: carboplatin alone, grey dashed curves: diverse drugs alone, grey curves: measured PCI of combinations, black curves: calculated PCI of combinations). The mean PCI values for all curves can be seen in I. (E–H) These distribution curves of PCI values for the different drugs alone, measured and calculated combinations with carboplatin were used to define resistance categories via the system described in Fig. 2A. The classification borders for measured and calculated are marked by a green (μ) and red line (μ − 1 SD). The measured and calculated PCI values of the combinations carboplatin and one of the other four drugs are plotted against each other. Individual data points are color-coded depending on the chemoresistance category of the patient for the two single drugs. (I) Table representing the mean PCI values for all curves.

Figure 4 New system is used to test other drug combinations (exceptions).

(A and B) 30 ovarian carcinoma samples were treated with carboplatin and docetaxel (A), 36 with carboplatin and gemcitabine (B) alone or in combination. PCI of the single drugs and the two combinations was measured with the CTR-Test. Additionally, the PCI of the combinations was calculated with the presented formula. The frequency distributions of PCI values of the different settings were plotted (black dashed curves: carboplatin alone, grey dashed curves: docetaxel or gemcitabine alone, grey curves: measured PCI of combinations, black curves: calculated PCI of combinations). The mean PCI values for all curves can be seen in E. (C and D) The distribution curves of PCI values for the two drugs alone, measured and calculated combinations with carboplatin, were used to define resistance categories via the system described in Fig. 2A. The classification borders for measured and calculated are marked by a green (μ) and red line (μ − 1 SD). The measured and calculated PCI values of the combinations carboplatin and one of the two other drugs are plotted against each other. Individual data points are color-coded depending on the chemoresistance category of the patient for the single drugs. (E) Table representing the mean PCI values for all curves.

In order to investigate if our test system is also functional in a carboplatin-independent system, we tested two other drug combinations, 5-fluorouracil—SN-38 (active form of the prodrug irinotecan) (Figs. 5A and 5C) and 5-fluorouracil—oxaliplatin (Figs. 5B and 5D). These two combinations are standard therapies for colon carcinoma. Therefore, additional to the ovarian carcinomas other tumor types were measured as well. For the combination 5-fluorouracil—SN-38 32 ovarian carcinoma, one melanoma, one small cell bronchial carcinoma, one non-small cell lung carcinoma, one mamma carcinoma and four colon carcinoma were used. For the combination 5-fluorouracil—oxaliplatin 31 ovarian carcinoma, one melanoma, one small cell bronchial carcinoma, one non-small cell lung carcinoma, one mamma carcinoma and two colon carcinoma were tested. Cell growth inhibition was determined by the CTR-Test. Due to the fact that from colon carcinoma samples only a limited number of cells can be isolated and the samples were needed for regular commercial testing we used only six left over colon carcinoma samples for testing the aforementioned two combinations. We also used ovarian carcinoma samples and other tumor types, which are left overs of commercial CTR-Tests performed in the lab. Data analysis was performed like for the carboplatin combinations with regard to frequency distributions and the correlation between calculated and measured PCI values (Fig. 5). The histograms of both combinations show a distribution of the different PCI curves of the single drugs, the calculated and the measured values, which differs from an ideal distribution (Figs. 5A and 5B). The data points for both the combination 5-fluorouracil—SN-38 and 5-fluorouracil—oxaliplatin show a good distribution in the SR range. However, the rest of the data points show a divergent pattern (Figs. 5C and 5D), though, the correlation coefficient and the resistance classification between calculated and measured values (Table 3) exhibit values in a good range.

Figure 5 New system is used to test other carboplatin-independent drug combinations.

(A and B) A total of 32 ovarian carcinoma, one melanoma, one small cell bronchial carcinoma, one non-small cell lung carcinoma, one mamma carcinoma and four colon carcinoma were treated with 5-fluorouracil and SN-38 (A); 31 ovarian carcinoma, one melanoma, one small cell bronchial carcinoma, one non-small cell lung carcinoma, one mamma carcinoma and two colon carcinoma were treated with 5-fluorouracil and oxaliplatin (B) alone or in combination. PCI of the single drugs and the two combinations was measured with the CTR-Test and the PCI of the two combinations was also calculated with the presented formula. The frequency distributions of PCI values of the different settings were plotted (black dashed curves: 5-fluorouracil alone, grey dashed curves: SN-38 or oxaliplatin alone, grey curves: measured PCI of combinations, black curves: calculated PCI of combinations). The mean PCI values for all curves can be seen in E. (C and D) The distribution curves of PCI values for the two drugs alone, measured and calculated combinations with 5-fluorouracil, were used to define resistance categories via the system described in Fig. 2A. The classification borders for measured and calculated are marked by a green (μ) and red line (μ − 1 SD). The measured and calculated PCI values of the combinations 5-fluorouracil and one of the two other drugs are plotted against each other. Individual data points are color-coded depending on the chemoresistance category of the patient for the single drugs. (E) Table representing the mean PCI values for all curves.

Table 3 Correlation coefficient (R) and resistance classification (calculated equal to measured values) of 5-fluorouracil combinations.

	5-Fluorouracil & SN-38	5-Fluorouracil & Oxaliplatin	
Correlation coefficient (R)	0.86	0.92	
Resistance classification: calculated = measured	65.7%	70.3%	
Data set size (n)	35	37	

To adequately compare calculated and measured PCI values, we generated Bland-Altman-Plots for the different combinations (Fig. S1). For the combinations carboplatin and paclitaxel (Fig. S1A), carboplatin and Caelyx (Fig. S1B), carboplatin and etoposide (Fig. S1D), carboplatin and topotecan (Fig. S1E) as well as 5-fluorouracil and SN-38 (Fig. S1H) at 3% to 7% on the average, the calculated values are higher than the measured values. Furthermore, 95% of the deviations between measured and calculated values lie between +27% and −14%, except for 5-fluorouracil and SN-38 where the values lie between +31% and −26%.

The combination carboplatin and doxorubicin (Fig. S1C) shows a very good agreement between measured and calculated values. At only 0.7% on the average, the calculated values are higher than the measured values. A total of 95% of the deviations between measured and calculated values lie between +16% and −15%.

In agreement with our other results, the combinations carboplatin and docetaxel (Fig. S1F) and carboplatin and gemcitabine (Fig. S1G) exhibit worse values. At 12% and 11% on the average, the calculated values are higher than the measured values. A total of 95% of the deviations between measured and calculated values lie between +35% and −10% or +33% and −11%, respectively.

For the combination 5-fluorouracil and oxaliplatin (Fig. S1I), at 3% on the average, the calculated values are lower than the measured values. This is in contrast to the other combinations where the calculated values are mostly higher than the measured ones. A total of 95% of the deviations between measured and calculated values lie between +48% and −53%.

Discussion

We developed a new system which allows to test the clinical relevance of drug combinations in vitro via the CTR-test and a formula based on an additive model. Our system uses tumor material from patients instead of cell lines and the drugs are applied in concentrations that are similar to or lower than the physiological maximal serum concentrations (Cmax) which leads to a high comparability to clinical data. Additionally, this system uses a quite simple mathematical model which is based on the concept of Loewe additivity (Loewe, 1953; Berenbaum, 1989). Our adapted concept states that one half of the ideal concentration of drug A could replace one half of the ideal concentration of drug B or vice versa. The adapted concept is employed to predict the efficacy of a combination by calculating its PCI value based on single drug measurements. In an additive situation, one half of the concentration of each drug applied in combination leads to an equal PCI of the drugs alone at full concentration (Figs. 1A and 1B).

To test our system, we used a set of 273 ovarian carcinoma samples and measured the resistance profiles of carboplatin and paclitaxel alone or in combination by the CTR-Test. The adapted concept was used to calculate both the combination concentration and its PCI value. The setting of ovarian carcinoma was chosen because it is standardly treated with carboplatin and paclitaxel in combination. The aim of this test was to verify the accuracy of calculated PCIs of the combination based on the single drug measurements. Therefore, calculated and measured PCI values of the combination were compared (Fig. 1D). Since the data exhibit a correlation coefficient of 0.84 (Table 2), our system shows to be highly functional in predicting the combination PCI values based on the single drug measurements in the case of carboplatin and paclitaxel. This high correlation coefficient indicates a close relationship between the conceived ideal case and the actual activity of the two drugs combined. This is proven by the similarity between the frequency distributions of the PCI values and the theoretical ideal distributions (Fig. 1C). Consequently, the combination of carboplatin and paclitaxel functions in an additive way and the corresponding PCI value can be calculated.

An important step for predicting the efficacy of a chemotherapy is the resistance classification of tested drugs and their combinations based on PCI values. In detail, the resistance categories are determined individually via the frequency distribution of carboplatin, paclitaxel and the measured and calculated combination (Fig. 2A). Due to the importance of a correct classification the conformity of the measured and the calculated based classification was tested. The calculated resistance classification of the combination carboplatin and paclitaxel is in almost 80% of the cases in agreement with the actually measured resistance category (Fig. 2B and Table 2). This leads to the assumption that it is feasible and sufficient to predict the chemoresistance of a tumor to a drug combination by measuring the single drugs. To test this assumption, the measured resistance category was compared with the underlying single drug resistance classification. The postulated assumption holds true when the two individual drugs both belong to the same resistance category in case of SR and ER. However, when the resistance categories differ or both are MR, the prediction of the resistance category of the combination is less precise (Fig. 2C).

This suggests that if a patient is to be treated with a combination of drugs and the single drug measurements result in the same resistance categories (SR, ER), the calculated resistance category most likely leads to a clinical benefit or to no clinical benefit of the patient, respectively. If both drugs are rated SR or ER, the patient can be treated with the corresponding combination or should get another combination, respectively. Taken together, our data suggest that the best combination is composed of the most efficient and the worst combination is composed of the least efficient single drugs. For MR cases it should be explored if there may be a more functional combination. In case of different resistance categories of the single drugs, it makes sense to also test the combination because there is no precise prediction possible about the efficacy of the combination.

Those conclusions are so far only based on the combination of carboplatin and paclitaxel. In order to prove our system in a wider range we tested other carboplatin combinations applied in ovarian cancer treatment (Fig. 3). Looking at the correlation coefficients of the combinations carboplatin and Caelyx, carboplatin and doxorubicin, carboplatin and etoposide as well as carboplatin and topotecan, confirms the correlation seen for the carboplatin –paclitaxel combination (Table 2). Furthermore, the frequency distributions show an almost ideal distribution (Figs. 3A–3D) and the calculated resistance classifications are in around 70% of the cases equal to the measured classifications (Table 2). Therefore, all previous assumptions could be confirmed by testing other carboplatin combinations.

Moreover, it was verified if our system is also functional in a carboplatin independent setting and for other tumor species. Thus, we examined the combinations of 5-fluorouracil with SN-38 and of 5-fluorouracil with oxaliplatin, which are standard therapies for colon carcinoma (Fig. 5). The data points for the combinations 5-fluorouracil and SN-38 as well as 5-fluorouracil and oxaliplatin exhibit a good distribution in the SR area in the plot (Figs. 5C and 5D). However, the remaining values show a worse distribution. In addition, their frequency distributions are distinct from an ideal distribution (Figs. 5A and 5B). Nevertheless, comparing their correlation coefficients and the accuracy of resistance classification to the diverse carboplatin combinations reveals that both 5-fluorouracil combinations lie in a similar range (Table 3). The high correlation coefficients are probably due to the good distribution of values in the SR area. These results are in agreement with the results of the different carboplatin combinations and support our previous assumptions.

In clinical practice drug combinations are often applied and therefore it is necessary to have an approach for testing drug combinations to provide the best treatment for individual patients. Regarding our results, produced by investigating different combinations and tumor types, such an approach could be offered by our presented system in a clinically relevant setting. In contrast to our system, other methods to test drug combinations are far away from testing in a clinically relevant way since they use cell lines instead of tumor samples (Edelman, Quam & Mullins, 2001; Kashif et al., 2015; Patra et al., 2016). However, our system still has to be validated by clinical data in order to prove its efficacy for a patient.

The basis of our system is an additivity concept. In case of the different tested combinations, which function in an additive way, it is sufficient to measure the single drugs and calculate the PCIs of the corresponding combinations as long as the single drugs both are classified either as SR or ER. Therefore it is possible to find the best or the worst combination, respectively. When single drugs are both classified as MR or belong to distinct resistance categories it is reasonable to measure the combination. Furthermore, if there is a priority for a specific combination as it is the case for colon carcinoma, which is by default treated either with 5-fluorouracil and SN-38 or 5-fluorouracil and oxaliplatin, the combination could be tested in the first place.

However, when tested combined drugs do not function in an additive way, this is resulting in possible limitations of our system and measurement of single drugs might not be sufficient. This effect was seen when we tested the combinations carboplatin and docetaxel as well as carboplatin and gemcitabine. These combinations exhibit frequency distributions which differ in a great extent from an ideal distribution (Figs. 4A and 4B). For example, the mean PCI values for gemcitabine and the combination with carboplatin are very high and the curves are shifted to the right (Figs. 4B and 4E). The data points in the plot are shifted to the right as well (Fig. 4D). A distinct distribution in the plot is seen for the data points of the combination carboplatin and docetaxel (Fig. 4C). In contrast to all other combinations, a data point from a single patient classified as SR for both drugs lies in the measured ER range. Additionally, the correlation coefficients and the accuracy of resistance classification are also worse than for the other tested combinations (Table 2). These discrepancies could be explained by a non-additive mode of action of the used combinations. In case of gemcitabine and carboplatin synergism is reported for cell line systems (Edelman, Quam & Mullins, 2001; De Brito Galvao et al., 2012; Jin et al., 2013; Tomita et al., 2014). The reason for the false classification in case of docetaxel could be an antagonistic effect, as shown before (Budman, Calabro & Kreis, 2002). Thus, calculating PCI values of a combination is not applicable in a non-additive situation since this might lead to false results. Therefore, if it is previously known that two drugs do not function in an additive way when applied in combination, the combination needs to be tested directly. Thereby, via measuring the combination it might be possible to predict the efficacy of the combination.

Our previous results and conclusions are perfectly supported by Bland-Altman-Plots (Fig. S1). For the non-additive combinations carboplatin and gemcitabine and carboplatin and docetaxel, which showed poor correlation coefficients and accuracy of resistance classification, the plots revealed also significant differences between calculated and measured values. All other combinations that showed good correlation coefficients and accuracy of resistance classification, performed well in the plots and showed a good agreement between calculated and measured values. For the combination carboplatin and doxorubicin there is almost no difference between calculated and measured values. The larger differences of the other additive combinations may be due to the fact that our system is only functional when the single drugs are both classified as ER or SR. In case of MR or divergent resistance categories, the classification is less precise and leads to discrepancies. Therefore, the results of the Bland-Altman-Plots suggest, that our new system, including the way of analysis, is functional in predicting the efficacy of additive drug combinations when single drugs are classified either as SR or ER.

Moreover, our new system could be employed to identify unknown non-additive combinations by comparing measured and calculated values. A low correlation coefficient, a low agreement between calculated and measured values as shown in a Bland-Altman-Plot would indicate a variation from an additive situation. A method to improve the identification of non-additive combinations would be to use only the calculated values for determining the resistance borders and not consider the borders determined by the measured values. In case of synergistic effects, the calculated border would classify more data points as SR than in the measured setting. With the other borders, extreme high PCI values could have been classified as extreme resistance even though it is quite likely the combination is able to produce an effect. The same is the case for antagonistic effects, more values would be classified as ER by the use of the calculated borders. Therefore, antagonistic combinations would not be used for treatment. These assumptions still have to be validated by clinical data.

Nevertheless a practical approach for measuring and evaluating the efficacy of any drug combination—even for unknown drug combinations—could be the following approach. Since the resistance borders (SR/MR and MR/ER) of the single drugs are known, the resistance borders of any drug combination can be computed by calculating the means of the resistance borders of the single drugs, which are part of a certain combination. That means that for the future only the resistance borders for single drugs need to be measured and the resistance borders of drug combinations can be calculated. This would be a tremendous reduction in complexity and effort, since the resistance borders of any drug combination do not need to be measured any more.

In general, one can conclude that our system might be able on the one hand to measure single drugs and calculate the efficacy of their combination in an additive situation if the single drugs are both categorized as SR or ER. Thereby, the best and worst combination can be found. These data are backed up with clinical data in which the best single drugs were used for therapy in clinical trial settings (Orr, Orr & Kern, 1999; Mehta et al., 2001; Holloway et al., 2002; Loizzi et al., 2003; d’Amato et al., 2009; Kim et al., 2009; Matsuo et al., 2009). On the other hand, it could be employed to measure drug combinations if necessary, for example in case of other resistance categories or if a specific combination is preferred. If feasible, the single drugs should be measured in addition to the combination when it is planned to treat a patient with a certain combination therapy. Thereby, it is possible to detect which of the single drugs is not functional in case of a non-responding combination. Therefore, this system could be applied in order to detect chemotherapy resistances of single patients and help to provide the best therapy option for the individual patient. Despite the usefulness of single drug efficacy measurements, in the future this system has to be further validated by clinical data to prove that it can further improve clinical benefit.

Additionally, the use of this test could be extended to predict treatment outcomes of treatment plans with more than two drugs involved. Therefore, the used monotherapy’s concentration of the single drugs measured in combination, could be divided by the number of used drugs in this multi-component treatment. The extension of this model would help to close the gap between clinic and diagnostic even further.

However, it cannot be guaranteed that a patient responds to a chemotherapy when he is sensitive to a specific chemotherapeutic or combination since this test system is used to detect resistances rather then to identify presumably effective chemotherapeutics. Whether a therapy is effective could for example be influenced by the mode of application of the chemotherapeutics. When drugs are not effective in vitro, the patient will not respond to the therapy, independent on the application mode. If a drug is effective in vitro but is not applied effectively enough, the application mode could affect the efficacy of the therapy. Moreover, an in vitro system is not able to mimic all resistance mechanisms in the body, like the detoxification capability of a patient for example. Thus, to further improve the individualized treatment of cancer patients one could combine our test system to detect resistances against chemotherapeutics with pharmacological and/or toxicological investigations.

Last but not least, this method is not limited to chemotherapeutic drugs, but can be used for any anti-cancer drug combinations—including new targeted drugs—which act directly on the tumor cells.

Supplemental Information

Figure S1 Bland-Altman-Plots for the different tested drug combinations

(A–I) Measured and calculated PCI values of the different carboplatin (A–G) and 5-fluorouracil (H and I) combinations were shown in a Bland-Altman-Plot to provide an adequate description of agreement.

Click here for additional data file.

Table S1 Standard deviations of applied drugs and their combinations used for classification into the three resistance categories

Click here for additional data file.

Data S1 Raw data showing PCI values

Click here for additional data file.

Additional Information and Declarations

Competing Interests

Author Contributions

Human Ethics

Patent Disclosures

Data Availability

Dr. Kischkel declares that he is shareholder of TherapySelect, which offers the CTR-Test as a commercial diagnostic service. A patent for the described measuring and interpretation of drug combinations is pending. Julia Eich, Carina Meyer, Paula Weidemüller, Jens Krapfl, Rauaa Yassin-Kelepir, Laura Job and Marius Fraefel are or have been employees of TherapySelect, Heidelberg.

Frank Christian Kischkel conceived and designed the experiments, performed the experiments, prepared figures and/or tables, reviewed drafts of the paper.

Julia Eich and Carina I. Meyer analyzed the data, wrote the paper, prepared figures and/or tables.

Paula Weidemüller analyzed the data, prepared figures and/or tables.

Jens Krapfl, Rauaa Yassin-Kelepir, Laura Job and Marius Fraefel performed the experiments.

Ioana Braicu contributed reagents/materials/analysis tools.

Annette Kopp-Schneider analyzed the data, reviewed drafts of the paper.

Jalid Sehouli and Rudy Leon De Wilde contributed reagents/materials/analysis tools, reviewed drafts of the paper.

The following information was supplied relating to ethical approvals (i.e., approving body and any reference numbers):

TherapySelect offers the commercial testing of drug efficacies for viable tumor samples. For this testing, viable tumor specimen is shipped to TherapySelect’s laboratory. Customers (patients) fill out a consent and order form. In this form there is a section in which the patients can choose whether left over material can be used for research purposes.

For all used samples patient’s consent forms exist, which allow further scientific investigations.

For this paper no ethical approval was requested, since human tissue was initially removed for commercially performed diagnostic purposes.

The following patent dependencies were disclosed by the authors:

A patent for the described measuring and interpretation of drug combinations is pending. The reference number for the German patent application is 10 2016 011 765.6.

The following information was supplied regarding data availability:

The raw data has been supplied as a Supplementary File.

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
