# Peer review of "New in vitro system to predict chemotherapeutic efficacy of drug combinations in fresh tumor samples"

_PeerJ, doi:10.7717/peerj.3030_

## Round 0.1 · original submission · Minor Revisions

The manuscript has now been examined by three reviewers. They have raised a few points, which are relatively minor and easily addressable.

Reviewer 1 ·

Basic reporting

1. Please add a statement in regard to the approval by the applicable ethics committee, if applicable.

2. Consider rewording lines 38-39 of the conclusion, see also lines 378-379.

3. Figure 3c: I would suggest a 2D graph for better interpretation.

Experimental design

1. Line 116, how is the percentage of tumor cells quantified? Please add the number of cells that were seeded per well.

2. Line 187: Were the recurrent ovarian cancer samples pretreated? If yes, would this not effect subsequent treatment efficacy in vitro?

Validity of the findings

1. The authors state some limitations to the approach such as when drugs do not have a similar resistance category. How would the authors discuss other issues such as multiple chemotherapy cycles or sequential application of drugs?

2. How would the results be translated to improved patient treatment, should issues of pharmacology and/or toxicology be considered?

3. How does the approach you use in the described experiments compare to the MixLow method proposed by Boik et al. (2008, Statistics in Medicine)?

Additional comments

The manuscript is well written and provides an interesting and simple approach to modeling combination therapy in vitro. To my knowledge this topic has only been investigated in vitro by a very small number of publications, aside from simulation studies. Given some limitations the approach the authors describe could be a basis for additional research in this area.

Reviewer 2 ·

Basic reporting

The article is written in clear, acceptable english. Nevertheless, it should be proof-read by a native speaker.
The introduction to the topic is sufficient, and citation of previous work is provided. The background of applied methods and calculations could be explained in more detail for easier access to the subject.
The structure of the article is well organized and background, explaining the translational aspect of the study, is provided.
Figures are of suficient quality, are helpful in understanding the subject and are described accordingly.
All appropriate raw data has been made available.

Experimental design

The aim of the study was to develop and initially validate an in vitro method for the determination of resistance to specific chemotherapeutic therapies in order to improve the individualized therapy of cancer, with a focus on ovarian cancer. In the clinical situation of ovarian cancer, the guidelines for therapy as well as molecular determinants of chemoresistance are poorly established, therefore the study presented is of importance for the progress in understanding chemoresistance in ovarian cancer as well as for the individualization of cancer therapy.

Methods presented are described and citations are provided. Theoretical assumptions and predictive calculations are explained and argumented. Background information for the reproduction of the results can be accessed, to my knowledge.

Statement of existance of patients`consent was provided.

Minor comments/questions:
- How is the tumor content (number of epithelial cancer cells / fibroblasts) taken in account when in vitro testing is performed?
- Are there any "quality controls" regarding the use of cell cultures for chemosensitivity testing, e.g. ratio maximum inhibition / untreated control? Number of cell cycles?
This questions aims at the ability of dissociated tumor cells to be adherent and grow in in vitro culture.

Validity of the findings

The data presented is sound and controlled. Availability of data is given and supports the thesis.
The validity of findings, as far as presented, is indicated. Nevertheless, final conclusions about the translational, and especially predictive aspects of the assay claimed, need to be further investigated and followed up. The long term survival of patients with individualized therapy based on the test and control groups receiving standard therapies will be needed.

·

Basic reporting

No comment.

Experimental design

No comment.

Validity of the findings

No comment.

Additional comments

In the submitted manuscript entitled “New in vitro system to predict chemotherapeutic efficacy of
drug combinations in fresh tumor samples” Kischkel et al. describe the design of a new system to measure and calculate clinical relevance of drug combinations. The evaluation of the predicting efficacy relies on the CTR-test and a formula based on an additive model. According to this additive model half of the ideal concentration of a given drug A can replace one half of the ideal concentration of a given drug B. The authors found that this assumption is true if the response to the individual drug is similar (e.g. both show either slight resistance or extreme resistance). If the two drugs show very different resistances (i.e. response to drug A is slight resistance and response to drug B is extreme resistance), or medium resistance then the prediction of the resistance category for the combination becomes inaccurate. The model is also accurate in the case of additive responses but becomes less accurate when the drugs studied have synergistic effects. In summary, as the concept of having a new approach to test drug combinations is very interesting and of high clinical relevance, this study is likely to prove very useful for the prediction of drug response.

The following minor points should be addressed by the authors:

• In figure 2, it is not clear how the resistance borders were defined which makes it difficult for the reader to follow the experimental procedures used throughout the manuscript. The authors should provide an example on how borders were calculated.
• In figure 1, the authors should provide the quantification of the H3-Thymidine uptake as a measurement of cell growth inhibition.
• Correlation coefficients should be included directly on the figure wherever possible, not just in the legend.

---

## Round 0.2 · accepted · Accept

The relatively minor concerns and suggestions that had been raised by the reviewers have been adequately addressed in the new, revised version of the manuscript.